# ROYAL SOCIETY
# OPEN SCIENCE

green chemistry/organic chemistry/synthetic chemistry

DABCO functionalized ionic liquids, bis-2-amino-5-arylidenethiazol-4-ones, Knoevenagel condensation, stereoselectivity, aqueous media

**Author for correspondence:**
Wael A. A. Arafa
e-mail: waa00@fayoum.edu.eg

This article has been edited by the Royal Society of Chemistry, including the commissioning, peer review process and editorial aspects up to the point of acceptance.

# New dicationic DABCO-based ionic liquids: a scalable metal-free one-pot synthesis of bis-2-amino-5-arylidenethiazol-4-ones

## Wael A. A. Arafa[1,2] and Asmaa K. Mourad[2]

[1]Chemistry Department, College of Science, Jouf University, PO Box 2014, Sakaka, Aljouf, Kingdom of Saudi Arabia
[2]Chemistry Department, Faculty of Science, Fayoum University, PO Box 63514, Fayoum City, Egypt

(iD) WAAA, 0000-0002-9288-4143

Herein, a novel DABCO-based dicationic ionic liquid (bis-DIL) was easily prepared from the sonication of DABCO with 1,3-dichloro-2-propanol and then characterized by several techniques. Thereafter, under the ultimate green conditions, the performance of the bis-DIL was examined for the sono-synthesis of a new library of bis-2-amino-5-arylidenethiazol-4-ones via one-pot pseudo-five-component Knoevenagel condensation reaction of appropriate dialdehydes, rhodanine and amines. This protocol is tolerant towards several mono- and dialdehydes, excellently high yielding and affording access to the desired products in a single step within a short reaction time. Compared with the conventional methodologies, the proposed method displayed several remarkable merits such as milder reaction conditions without any side product, green solvent media, recording well in a variety of green metrics and applicability in gram-scale production. The recyclability of the bis-DIL was also investigated with an average recovered yield of 97% for six sequential cycles without any significant loss of the activity.

## 1. Introduction

4-Thiazolidinones are a promising class of heterocyclic scaffolds in the field of pharmaceutical chemistry [1–4]. In particular, 2-amino-5-arylidene-1,3-thiazol-4(5*H*)-one derivatives exhibited potential and interesting broad medicinal activities, for example, antiviral, antibacterial, anti-inflammatory and cardiotonic [5–9]. In order to prepare such heterocyclic moiety, several synthetic strategies

**A**
[DABCO-EtOH][AcO]

**B**
[DABCO-DOL][X]
**X** = AcO, BF$_4$, PF$_6$

**Figure 1.** Structures of the [DABCO-EtOH][AcO] and [DABCO-DOL][X] ILs.

have been established [10–12], but these reported protocols necessitate multi-steps, elevated temperature, laborious work-up and long reaction times. Anderluh *et al*. [6] reported an acid-catalysed multi-component preparation of 2-amino-5-alkylidenethiazol-4-ones under microwave irradiation at high temperatures using an excess amount of aldehyde in order to obtain acceptable yields. Consequently, this methodology cannot be considered as a green method. Also, Shariati & Baharfar [13] reported a method for synthesizing 2-amino-5-arylidenethiazol-4-ones using MgO nanoparticles. Despite an acceptable yield being obtained, the reaction took more time to complete in addition to the hazards resulted from using metal oxides, especially when the product is directed to human use. In green chemistry, ionic liquids (ILs) have been identified as environmentally friendly dual reagents (catalyst and solvent) in respect of their notable properties, for example, low cost, availability, recyclability, capability to dissolve a broad scope of compounds in addition to their potential to be used in gram-scale production [14,15]. They have been introduced in several organic conversions, for example, Michael addition, Aldol reaction, Mannich reaction, borane reductions, Knoevenagel condensation and Friedel–Crafts alkylation reactions [16–18]. The dicationic ILs containing two head groups linked by a rigid or flexible spacer have been explored as a powerful catalyst for several transformations. This type of IL demonstrates unique merits compared to monocationic ILs [19,20]. In recent years, DABCO, as a cage-like molecule, has been used as a green catalyst for several organic conversions [21,22]. This cage-like structure makes the nitrogen lone pair precisely localized, and consequently, DABCO becomes more attuned for quaternization. Accordingly, this merit has been used for the assembly of several types of DABCO-based ILs [23–27]. Latterly, two new ILs based on DABCO were prepared by our group (figure 1) and used successfully as catalysts in the Claisen–Schmidt condensation [28] and multisubstituted imidazoles synthesis [29]. As one-pot multi-component reactions (MCRs) can lead to robust and green synthetic protocols for the rapid preparation of small drug-like compounds [30], the combination of both ILs and MCRs in a single strategy would lead to rapid assembly of highly substituted heterocyclic compounds with high potential applications in pharmaceutical chemistry [31]. Recently, many industrial and scientific investigations have relied mainly on the use of ultrasound-assisted multi-component reactions [32]. Sonochemical preparations have numerous advantages compared to conventional warming strategies such as straightforward reaction routes and enhanced reaction rates and yields [32]. During the sono-synthesis reaction, the energy required to improve the rate is supplied by cavitation in which, on collapsing, the generated bubbles inside the solution release a significant amount of energy in a short reaction time. Chemically, the cavitation impacts are basically reliant on the substances inside the collapsing bubbles and thus on the solvent nature [33]. As the ILs have low vapour pressure and high miscibility with water, they can control the chemicals inside the collapsing bubbles via reducing the vapour pressure of the solvent. Consequently, the reactant's compressional heating is elevated and that in turn improves the reaction rates [34]. Accordingly, development of an effective protocol for the preparation of thiazol-4(5*H*)-one cores sounds interesting from the medicinal and chemical standpoint. As a part of our interest in developing novel strategies for heterocycles assembly [28,29,35–37], herein we present the synthesis of a broad scope of mono- and bis-2-amino-5-arylidenethiazol-4-ones using new DABCO-based ILs under sustainable reaction conditions. In this context, recyclability and recoverability of the IL as well as the reaction mechanism were also investigated. To the best of our knowledge, this is the first report claiming new bis-2-amino-5-arylidenethiazol-4-ones (schemes 2–4).

## 2. Results and discussion

At the outset, we began our investigation with the synthesis of a novel category of DABCO-based ILs designated as [DABCO$_2$-C$_3$-OH]2X (bis-DILs). The multistep strategy for synthesizing novel bis-DIL

**Scheme 1.** Synthesis of bis-DILs (**3a,b**) and DIL (**4**).

**7a–d**
**7a**, R$_1$–R$_2$=–(CH$_2$)$_5$–
**b**, R$_1$–R$_2$=–(CH$_2$)$_2$–O–(CH$_2$)$_2$–
**c**, R$_1$–R$_2$=–(CH$_2$)$_4$–
**d**, R$_1$ = PhCH$_2$, R$_2$ = H

**Scheme 2.** Synthesis of derivatives **7a–d**.

catalysts is displayed in scheme 1. Under sonication (US), two moles of DABCO were quaternized with one mole of 2,3-dichloropropan-1-ol (**1**) in EtOH to afford the corresponding chloride salt **2** in 98% yield (scheme 1). Next, the chloride salt **2** underwent an anion-exchange reaction on reacting with NaOAc or NaBF$_4$ in ethanol to afford the required dense bis-DILs **3a,b** in excellent yields under ultrasound irradiation (scheme 1). The bis-DILs **3a,b** were easily miscible with H$_2$O, EtOH and CH$_3$CN while being immiscible with *n*-hexane. The chemical structure of the prepared bis-DILs **3a,b** was elucidated by IR, [1]H, [13]C, [19]F NMR and high-resolution mass spectra (HRMS). In [19]F NMR of compound **3b**, the singlet at $\delta$ −148.61 ppm established the presence of BF$^{4-}$ group (electronic supplementary material, S5). As outlined in scheme 1, the novel bis-DILs **3a,b** comprise two free tertiary amine units and one free hydroxyl group on the same molecule. These active sites can easily activate Knoevenagel condensation reactions (scheme 5).

To demonstrate our assumption, we initially investigated the efficiency of bis-DILs (**3a,b**) as powerful ILs in promoting the Knoevenagel condensation reaction of commercially obtainable rhodanine **5a** (2.0 mmol), piperidine (2.0 mmol) and 2-hydroxy-5-methylisophthalaldehyde **6a** (1.0 mmol) (scheme 2). To assess the methodology from a greener perspective, the ultrasound conditions were combined under regimes of aqueous solvent. Then, in order to obtain the optimum reaction conditions, several variables were thoroughly investigated such as energy source type, reaction time, IL loading, IL composition and solvent system. It was observed that, in the absence of IL, a poor yield of the hitherto unreported bis-2-(piperidin-1-yl)thiazol-4(5*H*)-one (**7a**) was observed at 40°C (table 1, entry 1). Whereas, when the model reaction proceeded using 2.0 equiv of bis-DIL **3a**, the reaction profile was very clean, derivative **7a** was obtained as the sole Knoevenagel condensation product (scheme 2), and no by-product was produced as was observed from the [1]H NMR spectrum of the crude product (table 1, entry 2).

A comparative result was observed on employing bis-DIL (**3b**) but with a slight decrease in the reaction rate (table 1, entry 3). The other ILs furnished lower yields of the required product **7a** in longer reaction times (table 1, entries 4 and 5) with the relative efficiency following the order bis-DIL[OAc] ≈ bis-DIL[BF$_4$] > bis-DIL[Cl] > DABCO.

**Table 1.** Preparation of **7a** using various conditions.[a]

| entry | additives | conditions | time (min) | yield (%) |
|---|---|---|---|---|
| 1 | none | US[b] | 120 | 17 |
| 2 | **3a**: bis-DIL[OAc] | US | 30 | 93 |
| 3 | **3b**: bis-DIL[BF$_4$] | US | 35 | 91 |
| 4 | **2**: bis-DIL[Cl] | US | 45 | 88 |
| 5 | DABCO | US | 120 | 67 |
| 6 | **4**: [DABCO-H][OAc] | US | 120 | 62 |
| 7 | **3a**: bis-DIL[OAc] | stirring | 200 | 86 |
| 8 | NP-ZnO | US | 120 | 55 |
| 9 | NP-TiO$_2$ | US | 120 | 67 |

[a]**5a** (2.0 mmol), **6a** (1.0 mmol), piperidine (2.0 mmol) and IL (2.0 equiv) in water (10.0 ml) at 40℃.
[b]Ultrasonic irradiation.

**Table 2.** Impact of bis-DIL[OAc] **3a** loading during the preparation of **7a**.[a]

| entry | bis-DIL[OAc] (equiv) | time (min) | yield (%) |
|---|---|---|---|
| 1 | 2.0 | 30 | 93 |
| 2 | 2.5 | 15 | 96 |
| 3 | 3.0 | 12 | 99 |
| 4 | 3.5 | 10 | 99 |
| 5 | 4.0 | 10 | 99 |
| 6 | 5.0 | 10 | 99 |

[a]The mixture of **5a** (2.0 mmol), **6a** (1.0 mmol), piperidine (2.0 mmol) and **3a** in water (10.0 ml) was sonicated at 40℃.

Subsequently, the important role of the ILs hydroxyl group in stimulating the carbonyl groups of both rhodanine and aldehydes was also studied to have a better understanding of the catalytic system under investigation [23,38]. To do this, [DABCO-H][OAc] (DIL) **4** was synthesized [28,38] (scheme 1) and employed for catalysing the model reaction (scheme 2) instead of bis-DILs **3a,b**. It was found that the obtained yield of **7a** was diminished to 62% (table 1, entry 6) which is much lower than the yield obtained by using bis-DILs (approx. 92%). The aforementioned observation affirmed the essential role of the ILs hydroxyl group in stimulating the carbonyl groups (scheme 5). Also, the reaction yield was decreased on performing the reaction using the promising bis-DIL[OAc] under stirring instead of ultrasonication (table 1, entry 7). Furthermore, the model reaction was examined employing another two catalysts, ZnO and TiO$_2$ nanoparticles (2.0 equiv; electronic supplementary material, S36–S39). The latter catalysts were found to be less effective and delivered the required product **7a** in only 55 and 67% yields, respectively (table 1, entries 8 and 9). Moreover, increasing the catalyst loading of TiO$_2$ to 3.0 equiv neither increases the reaction yield nor the reaction rate. Thereby, in the next investigations, the bis-DIL[OAc] **3a** will be used as the optimal choice for further studies under ultrasonic irradiation.

In continuation of our trials to promote the results, the effect of bis-DIL[OAc] loading on the aforementioned process (scheme 2) was also studied. Increasing the bis-DIL[OAc] loading to 2.5, 3.0 and 3.5 equiv enhanced both the reaction rate and the yield (table 2, entries 2–4). Nevertheless, it was observed that a higher bis-DIL[OAc] loading (4.0 and 5.0 equiv) did not enhance the reaction rate (table 2, entries 5 and 6).

In order to probe the impact of the solvent on the reaction rate, various solvents were screened to select the optimal one. It was found that the selection of the solvent had a considerable impact on this conversion (scheme 2). When the reaction was performed in a polar protic solvent like EtOH at different temperatures, the reaction yield was slightly decreased (table 3, entries 1 and 2). A further lowering in the yield was observed when the reaction was conducted in polar aprotic solvents such as CH$_2$Cl$_2$ (DCM) and MeCN (table 3, entries 3 and 4). Moreover, a moderate yield was obtained on employing solvent-free conditions (table 3, entry 6). It was obvious that H$_2$O is the appropriate solvent for this conversion as it almost

**Table 3.** The impact of temperature and solvents on the preparation of **7a**.[a]

| entry | solvent | temperature (°C) | yield (%) |
|---|---|---|---|
| 1 | EtOH | 40 | 94 |
| 2 | EtOH | 60 | 75 |
| 3 | DCM | 40 | 45 |
| 4 | MeCN | 30 | 49 |
| 5 | $H_2O$ | 40 | 99 |
| 6 | None | 40 | 77 |
| 7 | $H_2O$ | 23 | 67 |
| 8 | $H_2O$ | 30 | 97[b] |
| 9 | $H_2O$ | 60 | 72 |
| 10 | $H_2O$ | 70 | 55 |

[a]The mixture of **5a** (2.0 mmol), **6a** (1.0 mmol), piperidine (2.0 mmol) and **3a** (3.5 equiv) in solvent (10.0 ml) was sonicated for 10 min at different temperatures.
[b]20 min.

**Table 4.** The impact of amines on the preparation of **7a–d**.[a]

| entry | amine | mmol | time (min) | product | yield (%) |
|---|---|---|---|---|---|
| 1 | piperidine | 2.0 | 10 | **7a** | 99 |
| 2 | piperidine | 2.5 | 10 | **7a** | 93 |
| 3 | piperidine | 3.0 | 10 | **7a** | 85 |
| 4 | piperidine | 3.0 | 20 | **7a** | 82 |
| 5 | morpholine | 2.0 | 10 | **7b** | 93 |
| 6 | pyrrolidine | 2.0 | 30 | **7c** | 92 |
| 7 | benzyl amine | 2.0 | 17 | **7d** | 88 |

[a]The mixture of **5a** (2.0 mmol), **6a** (1.0 mmol), amine and **3a** (3.5 equiv) in water (10.0 ml) was sonicated at 40°C.

afforded full conversion (99%) in only 10 min (table 3, entry 5). These experimental results indicate that the present reaction (scheme 2) belongs to the 'on water' type and is thus eco-friendly [25].

To assess the effect of temperature on the reaction rate, the reaction was conducted at varied temperatures. Performing the model reaction (scheme 2) at ambient temperature (23°C) diminished the reaction yield (table 3, entry 7). While on lowering the reaction temperature from 40 to 30°C, the reaction gave access to derivative **7a** in comparable yield (97%) but in longer reaction time (table 3, entry 8). By continuously increasing the reaction temperature to 60 and 70°C, the reaction yields were dramatically decreased with the appearance of Cannizzaro product [12] as a side product (table 3, entries 9 and 10). The mass spectrum of this Cannizzaro product showed the characteristic peak at $m/z$ 248.1288 (M–H) corresponding to (2-hydroxy-3-(hydroxymethyl)-5-methylphenyl)(piperidin-1-yl)methanone (electronic supplementary material, S31). Also, the structure of this product was confirmed by [1]H NMR spectrum (electronic supplementary material, S30).

Furthermore, the influence of the molar ratios and the type of used amines on the above-mentioned model reaction was tested (table 4). As outlined in table 4, the isolated yields of **7a** differed considerably when varying the molar ratios of piperidine (entries 1–4). Equimolar amounts of piperidine and **5a** (2.0 mmol) were found to be enough to produce the desired product **7a** in excellent yield (table 4, entry 1). Unexpectedly, a large amount of piperidine (more than 2.0 mmol) was detrimental to the reaction yields (table 4, entries 2 and 3) even at longer time (table 4, entry 4). Accordingly, a plausible explanation for the latter behaviour could be that a competitive Cannizzaro reaction took place in which aldehydes were consumed under basic conditions [6]. Therefore, it is preferable to perform the model reaction (scheme 2) at low temperature (40°C) and using an exact molar ratio of amine. Likewise, the other amines provided the corresponding derivatives **7b–d** in good yields (table 4, entries 5–7).

**Scheme 3.** Ultrasound-assisted one-pot protocol for the preparation of bis-arylidenethiazolidinones **7a** and **7e–j**.

**Table 5.** The impact of US power intensity on the preparation of **7a**.[a]

| entry | power intensity (%) | Watt (W) | time (min) | yield (%) |
|---|---|---|---|---|
| 1 | 20 | 40 | 25 | 92 |
| 2 | 40 | 80 | 20 | 92 |
| 3 | 60 | 120 | 15 | 96 |
| 4 | 80 | 160 | 10 | 99 |
| 5 | 100 | 200 | 10 | 97 |

[a]The mixture of **5a** (2.0 mmol), **6a** (1.0 mmol), piperidine (2.0 mmol) and **3a** (3.5 equiv) in water (10.0 ml) was sonicated at 40°C.

Finally, the influence of the power intensity of ultrasonic irradiation on the aforesaid reaction (scheme 2) was investigated. The model reaction was screened at a variety of operating intensities, i.e. 20%, 40%, 60%, 80% and 100%, and the desired derivative **7a** was obtained in 92%, 92%, 96%, 99% and 97% yields, respectively (table 5, entries 1–5). Accordingly, the optimal power intensity was selected to be 80% as it gave excellent yield in a shorter reaction time (table 5, entry 4).

According to the above experiments (tables 1–5), sonication (80%) of **5a** (2.0 mmol), **6a** (1.0 mmol), piperidine (2.0 mmol) and bis-DIL (**3a**, 3.5 equiv) in water (10.0 ml) at 40°C for 10 min was specified as the optimum reaction condition in which derivative **7a** was isolated in 99% yield. Having successfully established an optimized strategy for the model reaction, we next embarked on the reaction scope and generality of the substrates. In the current investigation, a diverse range of functionalized dialdehydes (**6a–g**) underwent catalytic Knoevenagel condensation reaction expeditiously with rhodanine (**5a**) and piperidine to furnish the respective hitherto unreported bis-2-(piperidin-1-yl)thiazol-4(5H)-ones (**7a**, **7e–j**) in almost quantitative yields (up to 99%, scheme 3).

**Scheme 4.** Ultrasound-assisted one-pot protocol for the preparation of bis-arylidenethiazolidinones **7k–m**.

It was noted that the steric and electronic nature of substituents linked to the dialdehyde cores has a minor impact on the reaction efficiency. Impressively, the reaction was amenable to both the electron-donating group substituent on the dialdehyde counterpart, such as *p*-methyl (**7a**, 99%), and electron withdrawing groups like *p*-carboxylic (**7f**, 95%) and *p*-bromo (**7e**, 95%). Switching to sterically electron withdrawing partners, namely, **6d** and **6e**, also afforded the required products (**7g** and **7h**) in comparable yields (96% and 95%, respectively). Gratifyingly, 1*H*-pyrazole-3,5-dicarbaldehyde (**6f**) and 5,5′-methylenebis(2-hydroxybenzaldehyde) (**6g**) also underwent an efficient and smooth reaction with rhodanine to provide the condensed derivatives **7i** and **7j** in 96% and 97% yields, respectively. Furthermore, the generality of the thiazolidin-4-one moiety was also tolerated in the present reaction. The reaction smoothly proceeded with both 2-thioxothiazolidin-4-one (**5a**) and thiazolidine-2,4-dione (**5b**) as active methylene compounds. It was noted that rhodanine (**5a**) afforded slightly better yields of the corresponding products in comparison to derivative **5b**.

Furthermore, it is worth noting that both the reaction rate and yield were slightly affected by the steric nature of the dialdehydes. For example, terephthalaldehyde **6h** was smoothly involved in the reaction and afforded the desired product **7k** in excellent yield. Likewise, isophthalaldehyde **6i** provided the respective product **7l** in comparable yield (91%). Notwithstanding, phthalaldehyde **6j** relatively hampered the reaction and furnished the required product **7m** in 82% yield (scheme 4), which might be possibly due to steric factors.

Thereafter, the efficiency of bis-DIL (**3a**) to promote Knoevenagel condensation was compared with the previously reported methodologies in order to clear up the generality and feasibility of the present IL. A focused library of 2-amino-5-arylidenethiazol-4-ones (**9a–m**) was prepared with three-component reaction of rhodanine (**5a**, 1.0 mmol), piperidine (1.0 mmol) and various aromatic and heteroaromatic aldehydes (**8a–m**, 1.0 mmol) under the optimum conditions (table 6). In all examples, the desired products (**9a–m**) were obtained as the sole products without the isolation of any side products in better yields and faster reaction rates than other reported protocols (table 6).

Furthermore, it was essential that the effectiveness of the as-synthesized bis-DIL (**3a**) be compared with other ILs in order to clear up its accessibility and merits. For this reason, the synthesis of compound **9a** was selected as the model reaction and was compared in terms of reaction time and

**Table 6.** Preparation of 2-amino-5-arylidenethiazol-4-ones **9a–m**.[a]

| entry | R$_1$ | R$_2$ | R$_3$ | yield (%) | time (min) | mp (°C) | ref. |
|---|---|---|---|---|---|---|---|
| 1 | H | H | H | 99 (83) | 15 (20) | 215–216 (213–215) | [6] |
| 2 | H | H | Cl | 99 (89) | 15 (100) | 206–208 (207–209) | [13] |
| 3 | H | H | CH$_3$ | 98 (89) | 15 (109) | 154–156 (152–154) | [12] |
| 4 | H | H | OCH$_3$ | 99 (88) | 15 (12 h) | 200–201 (197–199) | [5] |
| 5 | Br | H | H | 97 (89) | 20 (100) | 165–168 (162–164) | [13] |
| 6 | H | H | CN | 98 (64) | 15 (20) | 193–195 (192) | [6] |
| 7 | –O–CH$_2$–O– | | H | 98 (80) | 20 (12 h) | 185–186 (184) | [5] |
| 8 | H | OCH$_3$ | OCH$_3$ | 97 (79) | 15 (122) | 198–199 (198–200) | [12] |
| 9 | H | H | NH$_2$ | 96 (88) | 15 (104) | 213–215 (212–214) | [12] |
| 10 | H | OCH$_3$ | OH | 98 (83) | 15 (139) | 193–195 (192–194) | [12] |
| 11 | iso-Nicotinaldehyde | | | 98 (96) | 10 (112) | 171–173 (172–174) | [12] |
| 12 | Thiophene-2-carbaldehyde | | | 96 (53) | 15 (20) | 203–205 (203) | [6] |
| 13 | 1H-Indole-3-carbaldehyde | | | 98 (90) | 10 (20) | 244–246 (245) | [6] |

[a]**5a** (1.0 mmol), **8** (1.0 mmol), piperidine (1.0 mmol) and **3a** (2.0 equiv) in water (10.0 ml) at 40°C.

yield with some commercially available ILs [39–41]. As shown in table 7, compound **9a** was obtained in a shorter time and higher yield when bis-DIL (**3a**) was used, clarifying its applicability as an adequate IL for the promotion of these MCRs.

The structure of all the newly prepared compounds **7a–m** was interpreted by IR, NMR spectroscopy, as well as HRMS. The above-mentioned preparations could afford two isomers, Z and E. However, [1]H NMR spectra displayed only one type of methine proton at around $\delta$ 7.9 ppm, at lower chemical shifts than those anticipated for the E-isomers. Therefore, we can infer that the current method proceeds stereoselectivity to furnish the more stable thermodynamic Z-isomer [42].

According to the obtained experimental results and previous reports [18,25,38], a tentative mechanistic pathway is displayed in scheme 5. Presumably, the active hydrogen of compound **5a** was removed by the nitrogen lone pair of the bis-DIL (**3a**); as a result, the corresponding carbanion **10** was generated. At the same time, the OH group of the bis-DIL enhanced the electrophilicity of the carbonyl carbon atom of aldehyde due to intermolecular hydrogen bonding, which in turn reacted with intermediate **10** to yield the Knoevenagel condensation product **11**. Posteriorly, piperidine underwent a nucleophilic attack on the thiocarbonyl carbon of compound **11** owing to its increased electrophilicity due to the hydroxyl group of bis-DIL to afford the non-isolable intermediate **12** that finally performed an elimination reaction to afford the isolable derivatives **7–9** (scheme 5).

Additionally, in order to evaluate the potential of scaling up the present strategy, MCR was proceeded on the gram scale under the optimized conditions that in turn provided the desired derivative **7a** in excellent yield (98%; scheme 6). Also, to evaluate the current process in terms of carbon efficiency (CE) and waste, diversified green metrics including CE, process mass intensity (PMI), E-factor (EF), reaction mass efficiency (RME) and atom economy (AE) have been studied (scheme 6) [43]. The higher environmental compatibility parameters, for example, smaller values of both PMI and EF in

**Table 7.** Comparison of reaction conditions, times, yields and yield economies for the preparation of derivative **9a** using several ILs.[a]

| entry | IL | time (min)[b] | yield (%) | yield economy (%) |
|---|---|---|---|---|
| 1 | [C₁C₄im]Br[c] | 38 | 92 | 2.42 |
| 2 | [C₁C₄im]PF₆[c] | 25 | 96 | 3.84 |
| 3 | [C₁C₄im]OAc[c] | 30 | 89 | 2.96 |
| 4 | [C₁C₁im]Me₂PO₄[d] | 25 | 97 | 3.88 |
| 5 | [C₁C₄pyrr]OTf[e] | 30 | 87 | 2.90 |
| 6 | [HOC₂NH₃]HCO₂[f] | 55 | 91 | 1.65 |
| 7 | bis-DIL (**3a**, this work) | 15 | 99 | 6.60 |

[a]The mixture of **5a** (1.0 mmol), **8a** (1.0 mmol), piperidine (1.0 mmol) and IL (2.0 equiv) in water (10.0 ml) was sonicated at 40°C.
[b]Optimum time.
[c][C₁C₄im] = 1-butyl-3-methylimidazolium.
[d][C₁C₁im] = 1,3-dimethylimidazolium.
[e][C₁C₄pyrr] = 1-butyl-1-methylpyrrolidinium.
[f]2-Hydroxyethylammonium formate.

**Scheme 5.** Tentative mechanism for the assembly of derivatives **7–9**.

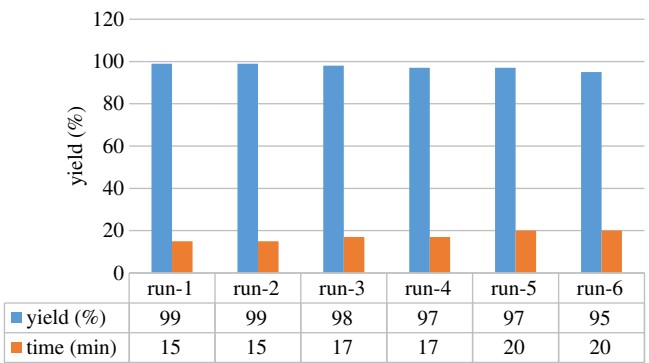

**Scheme 6.** Scaled-up preparation of derivative **7a**.

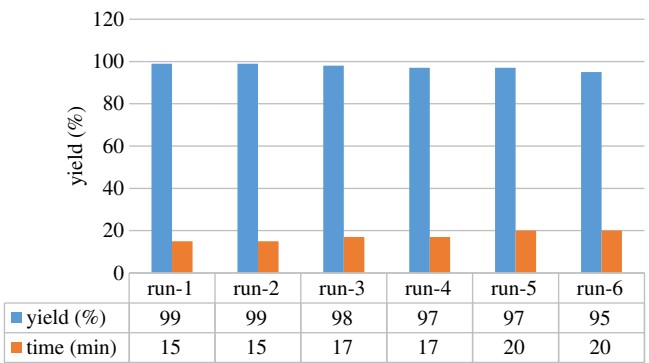

| | run-1 | run-2 | run-3 | run-4 | run-5 | run-6 |
|---|---|---|---|---|---|---|
| yield (%) | 99 | 99 | 98 | 97 | 97 | 95 |
| time (min) | 15 | 15 | 17 | 17 | 20 | 20 |

**Figure 2.** The evaluation of the reusability of the IL (**3a**) for the preparation of **7a**.

addition to high values of YE, CE, AE and RME confirmed the eco-friendly methodology of the developed protocol.

Finally, to test the possibility of bis-DIL (**3a**) recycling, the assembly of **7a** was chosen for this task. Under the optimized conditions, the recyclability of the bis-DIL (**3a**) was investigated by conducting the model reaction (scheme 2) repeatedly. After each cycle, the reaction product was isolated by filtration and the bis-DIL was recovered by the removal of $H_2O$ under reduced pressure then reused in the next cycle. Under the optimized reaction conditions, the bis-DIL was found to be very stable and can be reused up to six times without observing a remarkable loss of activity (figure 2). Moreover, the recycled IL after the sixth run was isolated from aqueous medium and characterized by $^1H$ NMR spectroscopy, which was the same as that of the original one.

## 3. Conclusion

In a promising approach, new DABCO-based dicationic ILs (bis-DILs) were synthesized and used for the first time as base ILs offering a considerable activity in Knoevenagel condensations at milder reaction conditions. Later, the developed bis-DILs were used to prepare a novel series of bis-2-amino-5-arylidenethiazol-4-ones by multi-component reaction of several aromatic dialdehydes, rhodanine and amines. The newly established strategy was successfully screened with other aromatic aldehydes and afforded the desired products in excellent yields. It was demonstrated that the presence of the hydroxyl group and two tertiary nitrogen motifs in the bis-DIL scaffold is crucial to increase the overall efficiency of the IL. Moreover, remarkable features of the developed protocol including high stereoselectivity, mild conditions, scalability, broad substrate scope, green solvent usage, chromatographic techniques avoidance, and step economy and AE would make this method attractive for green chemistry as well as organic synthesis. The developed bis-DILs could be recycled and

regenerated many times without any appreciable loss of effectiveness. Finally, several green metrics were studied and our protocol perfectly fitted in this grid.

# 4. Material and methods

Compounds **6a–g** were synthesized according to our previously reported method [44]. All chemicals were delivered and used without any further purification. $^1$H, $^{19}$F and $^{13}$C NMR spectra were recorded at 400 MHz, 377 MHz and 100 MHz, respectively. High-resolution mass spectra (HRMS) were recorded on a Bruker Daltonics microTOF spectrometer with an electrospray ionizer. FT-IR spectra were measured on a Perkin-Elmer Spectrum One spectrometer (KBr). Ultrasonication was done in a SY5200DH-T ultrasound cleaner. Melting points were measured by using the capillary tube method with an Electrothermal IA9100 apparatus (UK).

## 4.1. Synthesis of the catalysts bis-DIL (3a,b)

A mixture of DABCO (0.224 g, 2.0 mmol) and 2,3-dichloro-1-propanol (0.129 g, 1.0 mmol) in EtOH (15.0 ml) was sonicated for 1 h (TLC). The solvent was removed under reduced pressure to give derivative (**2**). Then, NaBF$_4$ (0.22 g, 2.0 mmol) or NaOAc (0.164 g, 2.0 mmol) was added to the residue of **2** in EtOH (15.0 ml). The formed solution was sonicated for 40 min, after which the solvent was removed under reduced pressure to give the corresponding ILs (**3a,b**) in almost quantitative yield.

### 4.1.1. Bis-DIL 3a

Yield 99%; colourless liquid; $^1$H NMR (400 MHz, CDCl$_3$): $\delta$ 3.93 (br, 1H, OH), 3.65 (br, 2H, C$H_2$), 3.54–3.51 (m, 6H, 3C$H_2$), 3.35–3.30 (m, 8H, 4C$H_2$), 3.20–3.13 (m, 13H, 6C$H_2$ + C$H$), 1.92 (s, 6H, 2C$H_3$); $^{13}$C NMR (101 MHz, CDCl$_3$): $\delta$ 175.0 (*C*=O), 69.9, 67.2, 64.5, 62.6, 61.9, 53.1, 52.9, 25.0 ppm (*C*H$_3$); IR (KBr, cm$^{-1}$): $\nu_{max}$ 3421 (br), 2923, 2880, 1638, 1115, 1053, 961; HRMS: $m/z$ [M]$^+$ calcd: 400.2685, found 400.2688; Anal. Calc. for C$_{19}$H$_{36}$N$_4$O$_5$: C, 56.98%; H, 9.06%; N, 13.99%. Found: C, 57.03%; H, 9.00%; N, 13.93%.

### 4.1.2. Bis-DIL 3b

Yield 99%; colourless liquid; $^1$H NMR (400 MHz, CDCl$_3$): $\delta$ 3.93–3.92 (br, 1H, OH), 3.80–3.78 (br, 2H, C$H_2$), 3.68–3.59 (m, 6H, 3C$H_2$), 3.48–3.36 (m, 8H, 4C$H_2$), 3.25–3.01 (m, 13H, 6C$H_2$ + C$H$); $^{13}$C NMR (101 MHz, CDCl$_3$): $\delta$ 70.0, 66.8, 64.2, 61.98, 61.90, 53.0, 52.9 ppm; $^{19}$F NMR (377 MHz, CDCl$_3$): $\delta$ −148.61 ppm; IR (KBr, cm$^{-1}$): $\nu_{max}$ 3402 (br), 2952, 2876, 1626, 1114, 1059, 960; HRMS: $m/z$ [M]$^+$ calcd: 456.2470, found 456.2477; Anal. Calc. for C$_{15}$H$_{30}$B$_2$F$_8$N$_4$O: C, 39.51%; H, 6.63%; N, 12.29%. Found: C, 39.56%; H, 6.58%; N, 12.24%.

## 4.2. Synthesis of derivatives 7a–j

A 25.0 ml round flask was charged with dialdehydes, **6a–j** (0.5 mmol), and/or aldehydes, **8a–m** (1.0 mmol), rhodanine (**5a**) and/or thiazolidine-2,4-dione (**5b**), amine (2.0 mmol in the case of **6a–j** and 1.0 mmol in the case of **8a–m**) and **3a** (3.5 equiv in the case of **6a–j** and 2.0 equiv in the case of **8a–m**) in 10.0 ml water. The reaction vessel was placed in the ultrasonic bath and sonicated at 40°C for an appropriate time. After completion of the reaction (confirmed by TLC, eluent: MeOH/DCM = 1 : 9 vol.), the solid that separated out was filtered and washed with H$_2$O, dried and recrystallized from a proper solvent to afford the analytically pure product. The catalyst was recovered from the aqueous layer under vacuum, washed with *n*-hexane, and reused for the next reactions. All known products (**9a–m**) were confirmed by comparing their melting points, IR, $^1$H NMR spectra and HRMS (see electronic supplementary material, S31-S36).

### 4.2.1. (5Z,5′Z)-5,5′-((2-hydroxy-5-methyl-1,3-phenylene)bis(methanylylidene))bis(2-(piperidin-1-yl)thiazol-4(5H)-one) 7a

Yield 99%; yellow solid; mp = 287–290°C; $^1$H NMR (400 MHz, DMSO-$d_6$): $\delta$ 11.50 (s, 1H, OH), 7.98 (s, 2H, =C$H$), 7.42 (s, 2H, Ar-H), 3.82–3.79 (m, 4H, piperidine-H), 3.72–3.71 (m, 4H, piperidine-H), 2.39 (s, 3H, C$H_3$), 1.70–1.65 (m, 6H, piperidine-H), 1.60–1.59 ppm (m, 6H, piperidine-H); $^{13}$C NMR (101 MHz,

DMSO-$d_6$): $\delta$ 180.4 (C=O), 173.9 (N=C), 153.3 (=CH), 132.8 (thiazole-C5), 148.4, 128.6, 128.0, 124.1 (Ar-C), 50.0, 49.4, 26.0, 25.3, 23.9 (piperidine-C), 21.3 ppm (CH$_3$); IR (KBr, cm$^{-1}$): $\nu_{max}$ 3421–2988 (OH), 1694 (C=O), 1604 (C=N), 1597 (C=C); HRMS: $m/z$ [M−H]$^+$ calcd: 495.1526, found 495.1523; Anal. Calc. for C$_{25}$H$_{28}$N$_4$O$_3$S$_2$: C, 60.46%; H, 5.68%; N, 11.28%. Found: C, 60.42%; H, 5.71%; N, 11.23%.

### 4.2.2. (5Z,5′Z)-5,5′-((2-hydroxy-5-methyl-1,3-phenylene)bis(methanylylidene))bis(2-morpholinothiazol-4(5H)-one) 7b

Yield 93%; yellow solid; mp = 214–217°C; $^1$H NMR (400 MHz, DMSO-$d_6$): $\delta$ 11.49 (s, 1H, OH), 7.98 (s, 2H, =CH), 7.44 (s, 2H, Ar-H), 4.07–4.02 (m, 4H, morpholine-H), 3.88–3.80 (m, 8H, morpholine-H), 3.70–3.64 (m, 4H, morpholine-H), 2.40 ppm (s, 3H, CH$_3$); IR (KBr, cm$^{-1}$): $\nu_{max}$ 3346–3129 (OH), 1686 (C=O), 1611 (C=N), 1589 (C=C); HRMS: $m/z$ [M−H]$^+$ calcd: 499.1090, found 499.1110; Anal. Calc. for C$_{23}$H$_{24}$N$_4$O$_5$S$_2$: C, 55.19%; H, 4.83%; N, 11.19%. Found: C, 55.22%; H, 4.79%; N, 11.09%.

### 4.2.3. (5Z,5′Z)-5,5′-((2-hydroxy-5-methyl-1,3-phenylene)bis(methanylylidene))bis(2-(pyrrolidin-1-yl)thiazol-4(5H)-one) 7c

Yield 92%; yellowish white solid; mp = 243–245°C; $^1$H NMR (400 MHz, DMSO-$d_6$): $\delta$ 11.29 (s, 1H, OH), 7.96 (s, 2H, =CH), 7.40 (s, 2H, Ar-H), 3.80–3.79 (t, J = 6.7 Hz, 4H, pyrrolidine-H), 3.51–3.50 (t, J = 6.4 Hz, 4H, pyrrolidine-H), 2.30 (s, 3H, CH$_3$), 2.06–2.01 ppm (m, 8H, pyrrolidine-H); IR (KBr, cm$^{-1}$): $\nu_{max}$ 3312–3151 (OH), 1698 (C=O), 1617 (C=N), 1576 (C=C); HRMS: $m/z$ [M−H]$^+$ calcd: 467.1211, found 467.1219; Anal. Calc. for C$_{23}$H$_{24}$N$_4$O$_3$S$_2$: C, 58.95%; H, 5.16%; N, 11.96%. Found: C, 58.90%; H, 5.21%; N, 11.89%.

### 4.2.4. (5Z,5′Z)-5,5′-((2-hydroxy-5-methyl-1,3-phenylene)bis(methanylylidene))bis(2-(benzylamino)thiazol-4(5H)-one) 7d

Yield 88%; yellow solid; mp = 267–269°C; $^1$H NMR (400 MHz, DMSO-$d_6$): $\delta$ 11.53 (s, 1H, OH), 10.08 (s, 2H, NH), 7.92 (s, 2H, =CH), 7.38–7.29 (m, 12H, Ar-H), 4.80 (s, 4H, CH$_2$), 2.29 ppm (s, 3H, CH$_3$); IR (KBr, cm$^{-1}$): $\nu_{max}$ 3384–3109 (OH), 1683 (C=O), 1610 (C=N), 1573 (C=C); HRMS: $m/z$ [M]$^+$ calcd: 540.1292, found 540.1288; Anal. Calc. for C$_{29}$H$_{24}$N$_4$O$_3$S$_2$: C, 64.43%; H, 4.47%; N, 10.36%. Found: C, 64.47%; H, 4.41%; N, 10.30%.

### 4.2.5. (5Z,5′Z)-5,5′-((5-bromo-2-hydroxy-1,3-phenylene)bis(methanylylidene))bis(2-(piperidin-1-yl)thiazol-4(5H)-one) 7e

Yield 95%; yellow solid; mp = 311–313°C; $^1$H NMR (400 MHz, DMSO-$d_6$): $\delta$ 11.56 (s, 1H, OH), 7.90 (s, 2H, =CH), 7.63 (s, 2H, Ar-H), 3.85–3.80 (m, 4H, piperidine-H), 3.60–3.55 (m, 4H, piperidine-H), 1.99–1.95 (m, 6H, piperidine-H), 1.77–1.71 ppm (m, 6H, piperidine-H); $^{13}$C NMR (101 MHz, DMSO-$d_6$): $\delta$ 180.0 (C=O), 174.2 (C=N), 152.2 (=CH), 133.2 (thiazole-C5), 154.9, 129.9, 126.5, 124.5 (Ar-C), 50.2, 49.7, 26.1, 25.4, 23.8 ppm (piperidine-C); IR (KBr, cm$^{-1}$): $\nu_{max}$ 3287–3021 (OH), 1684 (C=O), 1619 (C=N), 1612 (C=C); HRMS: $m/z$ [M]$^+$ calcd: 560.0550, found 560.0556; Anal. Calc. for C$_{24}$H$_{25}$BrN$_4$O$_3$S$_2$: C, 51.34%; H, 4.49%; N, 9.98%. Found: C, 51.33%; H, 4.45%; N, 9.94%.

### 4.2.6. 4-Hydroxy-3,5-bis((Z)-(4-oxo-2-(piperidin-1-yl)thiazol-5(4H)-ylidene)methyl)benzoic acid 7f

Yield 95%; yellow solid; mp = 307–309°C; $^1$H NMR (400 MHz, DMSO-$d_6$): $\delta$ 13.05 (br, 2H, 2OH), 8.10 (s, 2H, =CH), 7.67 (s, 2H, Ar-H), 3.89–3.88 (m, 4H, piperidine-H), 3.68–3.67 (m, 4H, piperidine-H), 1.89–1.85 (m, 6H, piperidine-H), 1.68–1.65 ppm (m, 6H, piperidine-H); $^{13}$C NMR (101 MHz, DMSO-$d_6$): $\delta$ 178.9 (C=O), 173.0 (C=N), 169.5 (C=O), 150.5 (=CH), 133.6 (thiazole-C5), 159.7, 130.6, 129.9, 125.4 (Ar-C), 50.2, 49.8, 25.7, 25.4, 23.4 ppm (piperidine-C); IR (KBr, cm$^{-1}$): $\nu_{max}$ 3390–2875 (OH), 1698, 1679 (C=O), 1605 (C=N), 1603 (C=C); HRMS: $m/z$ [M−H]$^+$ calcd: 525.1267, found 525.1262; Anal. Calc. for C$_{25}$H$_{26}$N$_4$O$_5$S$_2$: C, 57.02%; H, 4.98%; N, 10.64%. Found: C, 56.98%; H, 4.99%; N, 10.60%.

### 4.2.7. (5Z,5′Z)-5,5′-((4-hydroxy-[1,1′-biphenyl]-3,5-diyl)bis(methanylylidene))bis(2-(piperidin-1-yl)thiazol-4(5H)-one) 7g

Yield 96%; yellow solid; mp = 354–356°C; $^1$H NMR (400 MHz, DMSO-$d_6$): $\delta$ 11.40 (br, 1H, OH), 8.12 (s, 2H, =CH), 8.01 (s, 2H, Ar-H), 7.71–7.59 (m, 2H, Ar-H), 7.30–7.13 (m, 3H, Ar-H), 3.98–3.80 (m, 8H,

piperidine-H), 1.62–1.44 ppm (m, 12H piperidine-H); $^{13}$C NMR (101 MHz, DMSO-$d_6$): $\delta$ 179.5 (C=O), 171.7 (C=N), 151.2 (=CH), 132.9 (thiazole-C5), 163.1, 138.6, 136.2, 133.6, 130.8, 128.3, 126.5, 123.7 (Ar-C), 49.9, 49.1, 25.7, 25.4, 23.5 ppm (piperidine-C); IR (KBr, cm$^{-1}$): $\nu_{max}$ 3223–2998 (OH), 1698 (C=O), 1620 (C=N), 1611 (C=C); HRMS: $m/z$ [M]$^+$ calcd: 558.1761, found 558.1757; Anal. Calc. for $C_{30}H_{30}N_4O_3S_2$: C, 64.49%; H, 5.41%; N, 10.03%. Found: C, 64.52%; H, 5.39%; N, 9.97%.

### 4.2.8. (5Z,5′Z)-5,5′-((5-benzyl-2-hydroxy-1,3-phenylene)bis(methanylylidene))bis(2-(piperidin-1-yl)thiazol-4(5H)-one) 7h

Yield 95%; yellow solid; mp = 335–338°C; $^1$H NMR (400 MHz, DMSO-$d_6$): $\delta$ 11.52 (br, 1H, OH), 7.91 (s, 2H, =CH), 7.86 (s, 2H, Ar-H), 7.59–7.43 (m, 5H, Ar-H), 4.00 (s, 2H, CH$_2$), 3.10–3.08 (m, 8H, piperidine-H), 1.51–1.50 ppm (m, 12H, piperidine-H); $^{13}$C NMR (101 MHz, DMSO-$d_6$): $\delta$ 180.1 (C=O), 173.5 (C=N), 152.6 (=CH), 132.3 (thiazole-C5), 162.4, 139.3, 137.4, 133.7, 129.2, 128.7, 126.4, 123.8 (Ar-C), 49.6, 49.1, 25.7, 25.4, 23.5 (piperidine-C), 41.6 ppm (CH$_2$); IR (KBr, cm$^{-1}$): $\nu_{max}$ 3265–3098 (OH), 1689 (C=O), 1623 (C=N), 1617 (C=C); HRMS: $m/z$ [M + Na]$^+$ calcd: 595.1815, found 595.1811; Anal. Calc. for $C_{31}H_{32}N_4O_3S_2$: C, 65.01%; H, 5.63%; N, 9.78%. Found: C, 65.07%; H, 5.61%; N, 9.75%.

### 4.2.9. (5Z,5′Z)-5,5′-((1H-pyrazole-3,5-diyl)bis(methanylylidene))bis(2-(piperidin-1-yl)thiazol-4(5H)-one) 7i

Yield 96%; yellow solid; mp = 377–381°C; $^1$H NMR (400 MHz, DMSO-$d_6$): $\delta$ 14.03 (br, 1H, NH), 7.94 (s, 2H, =CH), 7.10 (s, 1H, pyrazole-H), 4.00–3.89 (m, 8H, piperidine-H), 1.78–1.69 ppm (m, 12H piperidine-H); $^{13}$C NMR (101 MHz, DMSO-$d_6$): $\delta$ 180.4 (C=O), 174.1 (C=N), 152.9 (=CH), 133.8 (thiazole-C5), 149.7, 109.8 (pyrazole-C), 49.7, 49.4, 25.4, 25.1, 23.7 ppm (piperidine-C); IR (KBr, cm$^{-1}$): $\nu_{max}$ 3245 (NH), 1701 (C=O), 1604 (C=N), 1596 (C=C); HRMS: $m/z$ [M−H]$^+$ calcd: 455.1322, found 455.1326; Anal. Calc. for $C_{21}H_{24}N_6O_2S_2$: C, 55.24%; H, 5.30%; N, 18.41%. Found: C, 55.21%; H, 5.33%; N, 18.38%.

### 4.2.10. (5Z,5′Z)-5,5′-((methylenebis(6-hydroxy-3,1-phenylene))bis(methanylylidene))bis(2-(piperidin-1-yl)thiazol-4(5H)-one) 7j

Yield 97%; yellow solid; mp = 365–368°C; $^1$H NMR (400 MHz, DMSO-$d_6$): $\delta$ 12.77 (br, 2H, 2OH), 7.99 (s, 2H, =CH), 7.30–7.29 (d, J = 7.7 Hz, 2H, Ar-H), 7.20 (s, 2H, Ar-H), 7.09–7.08 (d, J = 7.7 Hz, 2H, Ar-H), 4.10 (s, 2H, CH$_2$), 3.78–3.70 (m, 8H, piperidine-H), 1.79–1.59 ppm (m, 12H, piperidine-H); $^{13}$C NMR (101 MHz, DMSO-$d_6$): $\delta$ 179.7 (C=O), 173.2 (C=N), 153.5 (=CH), 132.3 (thiazole-C5), 161.2, 129.4, 129.0, 127.1, 126.7, 123.8 (Ar-C), 49.9, 49.7, 25.8, 25.5, 23.4 ppm (piperidine-C); IR (KBr, cm$^{-1}$): $\nu_{max}$ 3321–3075 (OH), 1697 (C=O), 1612 (C=N), 1592 (C=C); HRMS: $m/z$ [M−H]$^+$ calcd: 587.1786, found 587.1789; Anal. Calc. for $C_{31}H_{32}N_4O_4S_2$: C, 63.24%; H, 5.48%; N, 9.52%. Found: C, 63.25%; H, 5.44%; N, 9.48%.

### 4.2.11. (5Z,5′Z)-5,5′-(1,4-phenylenebis(methanylylidene))bis(2-(piperidin-1-yl)thiazol-4(5H)-one) 7k

Yield 98%; yellow solid; mp = 348–350°C; $^1$H NMR (400 MHz, DMSO-$d_6$): $\delta$ 7.91 (s, 2H, =CH), 7.21 (s, 4H, Ar-H), 3.65–3.61 (m, 8H, piperidine-H), 1.91–1.79 ppm (m, 12H, piperidine-H); $^{13}$C NMR (101 MHz, DMSO-$d_6$): $\delta$ 180.2 (C=O), 174.7 (C=N), 153.3 (=CH), 132.9 (thiazole-C5), 135.8, 130.2 (Ar-C), 49.9, 49.4, 25.5, 25.2, 23.1 ppm (piperidine-C); IR (KBr, cm-1): $\nu_{max}$ 1690 (C=O), 1622 (C=N), 1599 (C=C); HRMS: $m/z$ [M]$^+$ calcd: 466.1498, found 466.1495; Anal. Calc. for $C_{24}H_{26}N_4O_2S_2$: C, 61.78%; H, 5.62%; N, 12.01%. Found: C, 61.77%; H, 5.65%; N, 11.97%.

### 4.2.12. (5Z,5′Z)-5,5′-(1,3-phenylenebis(methanylylidene))bis(2-(piperidin-1-yl)thiazol-4(5H)-one) 7l

Yield 91%; yellow solid; mp = 359–362°C; $^1$H NMR (400 MHz, DMSO-$d_6$): $\delta$ 7.89 (s, 2H, =CH), 7.35–7.34 (d, J = 7.7 Hz, 2H, Ar-H), 7.09–7.06 (t, J = 7.9 Hz, 1H, Ar-H), 6.70 (s, 1H, Ar-H), 3.92–3.91 (d, J = 5.8 Hz, 5H, piperidine-H), 3.80 (br, 3H, piperidine-H), 1.88–1.72 ppm (m, 12H, piperidine-H); $^{13}$C NMR (101 MHz, DMSO-$d_6$): $\delta$ 180.0 (C=O), 174.4 (C=N), 153.4 (=CH), 132.5 (thiazole-C5), 135.2, 129.3, 128.8, 128.6, 124.3 (Ar-C), 50.1, 49.7, 25.4, 25.2, 23.2 ppm (piperidine-C); IR (KBr, cm$^{-1}$): $\nu_{max}$ 1699 (C=O), 1611 (C=N), 1589 (C=C); HRMS: $m/z$ [M]$^+$ calcd: 466.1498, found 466.1499; Anal. Calc. for $C_{24}H_{26}N_4O_2S_2$: C, 61.78%; H, 5.62%; N, 12.01%. Found: C, 61.75%; H, 5.67%; N, 11.96%.

## 4.2.13. (5Z,5′Z)-5,5′-(1,2-phenylenebis(methanylylidene))bis(2-(piperidin-1-yl)thiazol-4(5H)-one) 7m

Yield 82%; yellow solid; mp = 339–341°C; $^1$H NMR (400 MHz, DMSO-$d_6$): $\delta$ 8.10 (s, 2H, =C$H$), 7.79–7.70 (d, J = 7.9 Hz, 2H, Ar-H), 7.22–7.19 (t, J = 7.8 Hz, 2H, Ar-H), 3.90–3.83 (m, 4H, piperidine-H), 3.69–3.60 (m, 4H, piperidine-H), 1.88–1.80 (m, 6H, piperidine-H), 1.53–1.49 ppm (m, 6H, piperidine-H); $^{13}$C NMR (101 MHz, DMSO-$d_6$): $\delta$ 180.3 (C=O), 174.1 (C=N), 153.9 (=CH), 132.7 (thiazole-C5), 135.9, 130.1, 128.5 (Ar-C), 49.6, 49.2, 25.7, 25.3, 23.5 ppm (piperidine-C); IR (KBr, cm$^{-1}$): $\nu_{max}$ 1689 (C=O), 1621 (C=N), 1613 (C=C); HRMS: $m/z$ [M + Na]$^+$ calcd: 489.1396, found 489.1397; Anal. Calc. for $C_{24}H_{26}N_4O_2S_2$: C, 61.78%; H, 5.62%; N, 12.01%. Found: C, 61.79%; H, 5.60%; N, 11.98%.

Data accessibility. The datasets supporting this article have been uploaded as part of the electronic supplementary material.

Authors' contributions. W.A.A.A. designed and performed the experimental part of the work, managed the research, performed data analysis and wrote the manuscript. A.K.M. assisted with performing experimental part, data analysis, writing and revising of the manuscript.

Competing interests. We have no competing interests.

Funding. This study was financially supported by the Jouf University, project no. 634/39.

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
