## [Reviewer comments · Royal Society Open Science]

Review History

RSOS-190617.R0 (Original submission)

Review form: Reviewer 1

Is the manuscript scientifically sound in its present form?

No

Are the interpretations and conclusions justified by the results?

No

Is the language acceptable?

Yes

Is it clear how to access all supporting data?

Not Applicable

Do you have any ethical concerns with this paper?

No

Have you any concerns about statistical analyses in this paper?

No

Recommendation?

Reject

Comments to the Author(s)

Detailed remarks are given in the attached file (Appendix A).

Review form: Reviewer 2

Is the manuscript scientifically sound in its present form?

Yes

Are the interpretations and conclusions justified by the results?

Yes

Is the language acceptable?

Yes

Is it clear how to access all supporting data?

Yes

Do you have any ethical concerns with this paper?

No

Have you any concerns about statistical analyses in this paper?

No

Recommendation?

Accept as is

Comments to the Author(s)

The manuscript described the preparation of a DABCO based di-cationic ionic liquid and the application of this ionic liquid in a one-pot pseudo-five-component Knoevenagel condensation reaction. This work is an extension of studies previously published by the authors and appears to be carefully carried out. This work is publishable.

Review form: Reviewer 3

Is the manuscript scientifically sound in its present form?

No

Are the interpretations and conclusions justified by the results?

No

Is the language acceptable?

Yes

Is it clear how to access all supporting data?

Yes

Do you have any ethical concerns with this paper?

No

Have you any concerns about statistical analyses in this paper?

No

Recommendation?

Reject

Comments to the Author(s)

The results reported are incremental, not proving any significant novelty. Basic ionic liquids are known and their application for Knoevenagel reaction is well established. Indeed, the reaction is catalysed by common ILs even in absence of basic sites. The authors failed to demonstrate in what way is the catalyst (3) better than other IL's reported by other groups for the Knoevenagel condensation (i.e. ACS Sustainable Chem. Eng., 2018, 6 (11), pp 13676–13680, Synthetic Communications 2018 48(9), pp. 1060-1067, ChemistrySelect, 2018, 3(17), pp. 4745-4749, Applied Catalysis B: Environmental Vol. 223, 2018, Pages 228-233, New J. Chem., 2015, 39, 9132-9142 without being exhaustive). Other ILs may have not been used for the specific reaction in the paper, but the authors should compare with well stabilised benchmark reaction to establish the real potential of the system reported.

Regarding the green metric, are the solvents used for the recovery of the ILs taking into the account. The calculations should be provided not only the formulas to check the validity of the numbers.

The recycling of the ILs require the separation of the product, remove the water (energy) washing with ether and then reuse. This is far to be a green process both water and organic solvent waste are generated.

There are in the literature simpler and greener methods for the Knoevenagel reactions even under continuous flow conditions (RSC Adv., 2013,3, 23075-23079, 2018, Asian Journal of Organic Chemistry 7(10), pp. 2061-2064)

Decision letter (RSOS-190617.R0)

24-Apr-2019

Dear Professor Arafa:

Manuscript ID: RSOS-190617

Title: "New Di-cationic DABCO-based Ionic Liquids: Green and Scalable Metal-Free One-pot Synthesis of Bis-2-amino-5-arylidene-thiazol-4-ones"

Thank you for submitting the above manuscript to Royal Society Open Science. Your paper was sent to reviewers and their comments are included at the bottom of this letter.

In view of the concerns raised by the reviewers, the manuscript has been rejected in its current form. However, a new manuscript may be submitted which takes into consideration these comments.

Please note that resubmitting your manuscript does not guarantee eventual acceptance, and that your resubmission will be subject to peer review before a decision is made.

Your resubmitted manuscript should be submitted by 22-Oct-2019. If you are unable to submit by this date please contact the Editorial Office.

On behalf of the Subject Editor Professor Anthony Stace and the Associate Editor Dr Andrew Harned

REVIEWER(S) REPORTS:

Associate Editor Comments to Author ():

RSC Associate Editor:

Comments to the Author:

The reported work appears to have been carried out in a careful manner, but the reviewers have some significant concerns regarding significance and novelty of the work as a whole, and the true "greenness" of the procedure. I am not quite so critical of the significance/novelty as the nucleophiles used in these Knoevenagel reactions are rather "non-standard" compared to other ionic liquid-promoted Knoevenagel reactions.

I suggest the authors modify their manuscript in the following ways:

- (1) The authors have compared their new ionic liquid to nanoparticle catalysts, but there does not appear to be any direct comparison with other, more established, ionic liquids. This must be done.
- (2) They should reconsider the green chemistry message in the manuscript and modify as appropriate, keeping the reviewers' comments in mind.

(3) Clearly state the significance and novelty of the work with respect to the current state-of-the-art in the area.

In addition to the comments provided by the reviewers, I have a few suggested modifications of my own:

(1) To clarify Reviewer 1's comments about Scheme 2, the authors should define R1 and R2 for compounds 7a-d.

(2) Page 11, procedure titled "Synthesis of derivatives 7a-j": The authors state "The catalyst was recovered from the aqueous layer under vacuum, washed with diethyl ether and reused for the next reactions." I assume the "catalyst" here is the ionic liquid. While the ionic liquid may very well play an intimate role in the reaction, I have a hard time considering something that is used in 2-3.5 times excess as a "catalyst". It would be better to just refer to this as "ionic liquid" in all procedures.

(3) Edit the entire manuscript for grammar and word usage.

RSC Subject Editor:

Comments to the Author:

(There are no comments.)

Reviewers' Comments to Author:

Reviewer: 1

Comments to the Author(s)

Detailed remarks are given in the attached file.

Reviewer: 2

Comments to the Author(s)

The manuscript described the preparation of a DABCO based di-cationic ionic liquid and the application of this ionic liquid in a one-pot pseudo-five-component Knoevenagel condensation reaction. This work is an extension of studies previously published by the authors and appears to be carefully carried out. This work is publishable.

Reviewer: 3

Comments to the Author(s)

The results reported are incremental, not proving any significant novelty. Basic ionic liquids are known and their application for Knoevenagel reaction is well established. Indeed, the reaction is catalysed by common ILs even in absence of basic sites. The authors failed to demonstrate in what way is the catalyst (3) better than other IL's reported by other groups for the Knoevenagel condensation (i.e. ACS Sustainable Chem. Eng., 2018, 6 (11), pp 13676-13680, Synthetic Communications 2018 48(9), pp. 1060-1067, ChemistrySelect, 2018, 3(17), pp. 4745-4749, Applied Catalysis B: Environmental Vol. 223, 2018, Pages 228-233, New J. Chem., 2015, 39, 9132-9142 without being exhaustive). Other ILs may have not been used for the specific reaction in the paper, but the authors should compare with well stabilised benchmark reaction to establish the real potential of the system reported.

Regarding the green metric, are the solvents used for the recovery of the ILs taken into the account. The calculations should be provided not only the formulas to check the validity of the numbers.

The recycling of the ILs require the separation of the product, remove the water (energy) washing with ether and then reuse. This is far to be a green process both water and organic solvent waste are generated.

There are in the literature simpler and greener methods for the Knoevenagel reactions even under continuous flow conditions (RSC Adv., 2013,3, 23075-23079, 2018, Asian Journal of Organic Chemistry 7(10), pp. 2061-2064)

Author's Response to Decision Letter for (RSOS-190617.R0)

Appendix B.

Decision letter (RSOS-190827.R0)

22-May-2019

Dear Professor Arafa:

Title: New Di-cationic DABCO-based Ionic Liquids: A Scalable Metal-Free One-pot Synthesis of Bis-2-amino-5-arylidene-thiazol-4-ones

Manuscript ID: RSOS-190827

Thank you for submitting the above manuscript to Royal Society Open Science. Your manuscript has now been reviewed. The comments from reviewers are included at the bottom of this letter.

In view of the criticisms of the reviewers, the manuscript has been rejected in its current form. However, a new manuscript may be submitted which takes into consideration these comments.

Please note that resubmitting your manuscript does not guarantee eventual acceptance, and that your resubmission will be subject to peer review before a decision is made.

Your resubmitted manuscript should be submitted by 19-Nov-2019. If you are unable to submit by this date please contact the Editorial Office.

Sincerely,
Dr Laura Smith
Publishing Editor, Journals

On behalf of the Subject Editor Professor Anthony Stace and the Associate Editor Dr Andrew Harned.

RSC Associate Editor

Comments to the Author:

Both the Editor and one of the reviewers asked the authors to provide a comparison between the new ionic liquids discussed here and existing ionic liquids. The authors did not provide this comparison. Instead, they have said that because these are new transformations, existing comparisons in the literature are not present. This may be true, but does not address the concerns raised. At least one reaction should be attempted using several existing (commercially available) ionic liquids for comparison.

Reviewers' Comments to Author:

Author's Response to Decision Letter for (RSOS-190827.R0)

See Appendix C.

Decision letter (RSOS-190997.R0)

24-Jun-2019

Dear Professor Arafa:

Title: New Di-cationic DABCO-based Ionic Liquids: A Scalable Metal-Free One-pot Synthesis of Bis-2-amino-5-arylidene-thiazol-4-ones
Manuscript ID: RSOS-190997

It is a pleasure to accept your manuscript in its current form for publication in Royal Society Open Science. The chemistry content of Royal Society Open Science is published in collaboration with the Royal Society of Chemistry.

On behalf of the Subject Editor Professor Anthony Stace and the Associate Editor Dr Andrew Harned.

RSC Associate Editor

Comments to the Author:

The authors have included comparison experiments using several, more standard, ionic liquids. The designer ionic liquids reported in this manuscript appear to offer some modest advantage with respect to yield and reaction time. Nevertheless, other researchers may find these ionic liquids useful for other applications. Now that these results are included, I can recommend publication.

Reviewer(s)' Comments to Author:

Appendix A

Review for RSOS-190617

The paper describes a new method of synthesis of a series of 2-amino-5-arylidene-thiazol-4-one derivatives. According to the authors' declaration, the developed method is "green" due to the use of ionic liquid both as a catalyst and solvent and application of US. In my opinion, the use of term "green" in this case is unauthorized. According to the latest state of knowledge, most ionic liquids are not "green" and even considered as harmful due to their ecotoxicity. They are also not „cheap” as authors suggest in the introduction. This issue must be corrected. DABCO, like most of tertiary amines, is prone to quaternization and this cannot be considered unusual. Scheme 2 requires corrections, which must clearly show the full range of synthesis performed and be correlated with Table 1 and 4. Proposed reaction mechanism presented on Scheme 5, in the context of the description given in text, raises serious doubts. According to Scheme 5, the effectiveness of gemini ILs is determined by the effect of tertiary nitrogen atom and the free OH group in the bridge. The same centers are also present in a simple ionic liquid with one DABCO molecule, which however, was characterized by much lower activity. In this work, this issue has not been sufficiently clarified. In summary, the work should be assessed as a routine with weak potential. The developed method uses a significant amount of ionic liquid compared to main chemicals, which makes the method only of preparative importance. In my opinion, this manuscript, after necessary correction, qualifies for specialist journals focused on heterocyclic chemistry

Appendix B

Entry	Reviewers	Comments	Responses
1.	Editor	The authors have compared their new ionic liquid to nanoparticle catalysts, but there does not appear to be any direct comparison with other, more established, ionic liquids. This must be done.	By careful survey, we found that this is the first time for the reactions under investigation to be performed using ILs. We believe that offering a new chemical approach to perform our reactions makes our work significant and novel. Therefore, it is not possible to make this direct comparison as we offer here, in our manuscript, the first reported example of using ILs to catalyze these reactions and almost all the reported methods before only used nanoparticle catalysts. Accordingly, and due to the above-mentioned reasons we found it more convenient to remove the comparison presented in (Table 7) to avoid any confusion.
2.		They should reconsider the green chemistry message in the manuscript and modify as appropriate, keeping the reviewers' comments in mind.	Done; the manuscript as a whole was revised and the green chemistry message in our manuscript was reconsidered.
3.		Clearly state the significance and novelty of the work with respect the current state-of-the-art in the area.	The manuscript described the synthesis of hitherto unreported ILs and bis-thiazol-4-ones under ultrasonic conditions. We believe that these three elements are sufficient to prove the significance and novelty of the current work.
4.		To clarify Reviewer 1's comments about Scheme 2, the authors should define R1 and R2 for compounds 7a-d.	Done; both R1 and R2 are identified in Scheme 2.
5.		Page 11, procedure titled "Synthesis of derivatives 7a-j": The authors state "The catalyst was recovered from the aqueous layer under vacuum, washed with diethyl ether and reused for the	Done in this sentence and also in the manuscript as a whole.

		next reactions." I assume the "catalyst" here is the ionic liquid. While the ionic liquid may very well play an intimate role in the reaction, I have a hard time considering something that is used in 2-3.5 times excess a "catalyst". It would be better to just refer to this as "ionic liquid" in all procedures.	
6.		Edit the entire manuscript for grammar and word usage.	Done.
7.	Reviewer: 1	In my opinion, the use of term "green" in this case is unauthorized. According to the latest state of knowledge, most ionic liquids are not "green" and even considered as harmful due to their ecotoxicity. They are also not „cheap” as authors suggest in the introduction. This issue must be corrected.	Done; 1) "green" has been omitted. 2) "cheap" has been omitted.
8.		Scheme 2 requires corrections, which must clearly show the full range of synthesis performed and be correlated with Table 1 and 4.	Done; both R1 and R2 are identified in Scheme 2.
9.		Proposed reaction mechanism presented on Scheme 5, in the context of the description given in text, raises serious doubts. According to Scheme 5, the effectiveness of gemini ILs is determined by the effect of tertiary nitrogen atom and the free OH group in the bridge. The same centers are also present in a simple ionic liquid with one DABCO molecule, which however, was characterized by much lower activity. In this work, this issue has not been sufficiently clarified.	The synthesized bis-DIL, as described in the manuscript, have two free tertiary amine units. These two moieties capable of enhancing the reaction more than the IL containing only one tertiary amine unit. This fact was established in performing the reaction using IL derivative 4; in this case, both reaction yield and rate are diminished (Table1, entry 6).
10.	Reviewer: 3	The authors failed to demonstrate in what way is the catalyst (3) better than other IL's reported by other groups for the Knoevenagel condensation (i.e. ACS Sustainable Chem. Eng., 2018, 6 (11), pp 13676–13680, Synthetic Communications 2018 48(9), pp. 1060-1067, ChemistrySelect, 2018, 3(17), pp. 4745-4749,	Although, all of these papers described the Knoevenagel condensation reactions, none of them (or others) described the same reactions outlined in our manuscript. And hence, we cannot establish a fair comparison between our reactions and the reported reactions.

		Applied Catalysis B: Environmental Vol. 223, 2018, Pages 228-233, New J. Chem., 2015, 39, 9132-9142 without being exhaustive). Other ILs may have not been used for the specific reaction in the paper, but the authors should compare with well stabilized benchmark reaction to establish the real potential of the system reported.	
11.		Regarding the green metric, are the solvents used for the recovery of the ILs taking into the account. The calculations should be provided not only the formulas to check the validity of the numbers.	The mathematic equations that selected for the green metrics calculations (Chem. Soc. Rev. , 2012, 41 , 1485) did not take into account the solvents used. As described in the text, these calculations were done in order to evaluate the current process in terms of carbon efficiency and waste (the energy term was omitted).
12.		The recycling of the ILs require the separation of the product, remove the water (energy) washing with ether and then reuse. This is far to be a green process both water and organic solvent waste are generated.	Done; the manuscript as a whole was revised and the phrases related to green chemistry were modified.

Appendix C

Comments	Responses
Both the Editor and one of the reviewers asked the authors to provide a comparison between the new ionic liquids discussed here and existing ionic liquids. The authors did not provide this comparison. Instead, they have said that because these are new transformations, existing comparisons in the literature are not present. This may be true, but does not address the concerns raised. At least one reaction should be attempted using several existing (commercially available) ionic liquids for comparison.	The reaction related to the preparation of derivative 9a has been performed using several commercially available ionic liquids (6 examples) and the results were tabulated in Table 7. An appropriate paragraph has been also inserted along with supported references (Ref. No. 40-42). Highlighted in yellow